# Clinical and Molecular Differences Suggest Different Responses to Immune Checkpoint Inhibitors in Microsatellite-Stable Solid Tumors with High Tumor Mutational Burden [note 1]

**DOI:** 10.3390/cancers17162673

**Published:** 2025-08-16

**Authors:** Imran Nizamuddin, Tarik Demir, Katrina Dobinda, Ruohui Chen, Masha Kocherginsky, Peter Doukas, Neelima Katam, Carolyn Moloney, Devalingam Mahalingam

**Affiliations:** 1Robert H. Lurie Comprehensive Cancer Center, Feinberg School of Medicine, Northwestern University, Chicago, IL 60611, USA; inizamuddin@wustl.edu (I.N.); peter.doukas@nm.org (P.D.); neelima.katam@northwestern.edu (N.K.); 2Division of Hematology and Oncology, Developmental Therapeutics Institute, Northwestern University, Chicago, IL 60611, USA; tarik.demir@nm.org (T.D.); carolyn.moloney@nm.org (C.M.); 3Preventive Medicine, Feinberg School of Medicine, Northwestern University, Chicago, IL 60611, USA; katrina.dobinda@northwestern.edu (K.D.); ruohui.chen@northwestern.edu (R.C.); mkocherg@northwestern.edu (M.K.)

**Keywords:** genetic mutations, immune checkpoint inhibitors, liver metastasis, predictive biomarkers, solid tumors, tumor mutational burden

## Abstract

This study investigated the predictors of response to immune checkpoint inhibitors (ICIs) in 117 patients diagnosed with advanced solid tumors exhibiting a high tumor mutational burden (TMB) of ≥10 mutations per megabase (mut/Mb). Enhanced treatment responses were correlated with the absence of liver metastasis, a lack of prior systemic therapy, and the presence of mutations in the *TERT* gene. It is noteworthy that a TMB of ≥15 mut/Mb specifically correlated with improved outcomes in patients whose tumor types are not typically sensitive to ICIs. In contrast, mutations in the *MYC* pathway and the *MLL2* gene were associated with diminished treatment responses. Furthermore, demographic factors such as age, sex, and PD-L1 status did not exhibit significant predictive value. The findings underscore the validity of TMB as a biomarker, indicating that its effectiveness is influenced by tumor type, the presence of liver metastases, and specific genetic mutations.

## 1. Introduction

Over a decade ago, the Food and Drug Administration (FDA) approved the first immune checkpoint inhibitor (ICI), ipilimumab [1]. Since then, ICIs have transformed the treatment landscape for various cancers, with the capacity to elicit durable responses in tumors previously deemed untreatable [2]. Currently, the FDA has approved eleven ICIs for the treatment of over 20 different types of cancer, with ongoing research into additional indications [3].

A critical consideration is the identification of biomarkers to predict patients who would benefit from these therapies [4,5]. Increased programmed cell death ligand 1 (PD-L1) expression has been shown to result in increased response rates to ICIs depending on tumor type and expression level [6]. Tumors with high microsatellite instability (MSI-H), indicative of a deficient mismatch repair (dMMR) system, have also demonstrated sensitivity to ICIs [7]. In May 2017, the FDA granted accelerated approval of pembrolizumab for treatment of MSI-H/dMMR solid tumors, regardless of primary site, based on data from patients enrolled across five clinical trials [8]. This marked the first tumor-agnostic predictive biomarker approval for ICIs.

Recent interest has emerged in TMB as a predictive biomarker. TMB is defined as the total number of somatic mutations per coding area of a tumor genome [9]. It is hypothesized that highly mutated tumors, such as those with high TMB, may generate immunogenic neoantigens, thereby increasing T-cell reactivity [10]. While most MSI-H/dMMR tumors are TMB-high, not all TMB-high tumors exhibit MSI-H/dMMR [6]. Consequently, TMB has been investigated as a predictive biomarker even in the absence of MSI-H/dMMR.

Numerous retrospective studies have demonstrated a correlation between high tumor mutational burden (TMB) and enhanced response rates to specific ICIs, including nivolumab, pembrolizumab, and ipilimumab [11,12,13]. Large-scale meta-analyses have further substantiated these findings, indicating that patients with high TMB exhibit improved overall survival (OS) and progression-free survival (PFS) when treated with ICIs compared to those receiving chemotherapy alone [14,15]. A prospective exploratory analysis of the KEYNOTE-158 study investigated the association between TMB and response to pembrolizumab, revealing that patients with TMB ≥ 10 mut/Mb had a significantly higher likelihood response, irrespective of tumor type, with an overall response rate (ORR) of 29% [16]. Consequently, the FDA granted accelerated approval of pembrolizumab for the treatment of unresectable or metastatic tumors with high TMB that have progressed on prior therapies [17]. Nevertheless, this approval has sparked considerable debate within the oncology community [18,19]. While some view it as a means to provide additional therapeutic options for a population lacking effective alternatives, others have criticized it based on arbitrary TMB cutoffs, absence of clear improvement in OS, limited sample sizes in rare tumors, and questionable cost-effectiveness [20,21,22].

Subsequent studies have continued to explore the role of TMB in predicting response to ICIs. For instance, a study in advanced colorectal cancer found that only patients with MSI-H/dMMR or specific mutations exhibited improved OS when treated with ICIs [23]. Moreover, a retrospective study involving over 10,000 patients across various cancer types failed to identify a consistent TMB threshold predictive of response to ICIs in a tumor-agnostic manner [24].

Given these developments, there is interest in reassessing the current FDA approval criteria based on TMB and identifying specific patient populations and molecular phenotypes that may substantially benefit from ICIs.

## 2. Materials and Methods

### 2.1. Patients

Patients treated at the Robert H. Lurie Comprehensive Cancer Center of Northwestern University (Chicago, IL, USA) between 1 January 2015 and 31 December 2020 were identified using an Enterprise Data Warehouse grant. Eligible participants were individuals aged ≥18 years with biopsy-proven advanced (unresectable or metastatic) tumors treated with ICIs alone and high TMB (defined as ≥10 mut/Mb) identified from an evaluable tissue sample for biomarker analysis. ICIs included those targeting the CTLA-4, PD-1, and PD-L1 pathways. The study excluded tumors other than carcinomas and those with MSI-H, as well as patients with active autoimmune diseases requiring systemic treatment or who had received investigational therapy within four weeks prior to the administration of ICIs. Demographic and clinical characteristics, disease outcomes, and toxicity outcomes were retrospectively collected. PD-L1 expression was evaluated differently depending on type of cancer, with some cancers, such as non-small cell lung cancer, utilizing tumor proportion score (TPS) testing, and others, such as gastrointestinal cancers, utilizing combined positive score (CPS) testing. For both TPS and CPS, PD-L1 positivity was defined as expression level ≥ 1%.

Patients were categorized into two distinct cohorts based on their sensitivity to ICI treatments and FDA-approved indications. Group 1 represented tumors for which ICI treatments have FDA approval for standalone use, including melanoma, non-small cell lung cancer (NSCLC), adrenal corticoid carcinoma (ACC), squamous cell carcinoma (SCC) (anal, esophageal, skin, and head and neck), renal cell carcinoma, and urothelial carcinoma. Group 2 included patients with non-ICI-sensitive tumors (those for whom ICI treatments have not been approved by the FDA for standalone use), including breast cancer (BC), colorectal cancer (CRC), non-SCC gynecologic cancers (GC), pancreaticobiliary cancers (PBC), small cell lung cancer (SCLC), and upper gastrointestinal cancers (UGC) (esophageal, stomach, and duodenal adenocarcinomas).

### 2.2. Outcome Evaluation

The primary efficacy endpoint was ORR. Radiologic response to ICIs was assessed using iRECIST criteria [25] through independent central review based on computed tomography or magnetic resonance imaging. Target lesions and non-target lesions were defined at baseline to ascertain the overall tumor burden. Subsequent response evaluations were carried out through standard radiologic assessments at regular intervals, typically every 6–12 weeks. This process involved the quantitative measurement of target lesions and the qualitative assessment of non-target lesions. Patients were classified as responders (complete response (CR), partial response (PR), or stable disease (SD) ≥ 6 months) or non-responders (SD < 6 months or progression of disease (PD)). Additional endpoints included PFS and OS. PFS was defined as the time from ICI initiation to disease progression, recurrence, or death from any cause. Surviving patients without progression or recurrence were censored at time of last follow-up. OS was defined as the time from ICI initiation to death or last follow-up.

### 2.3. Safety

The assessment of toxicity outcomes was based on the presence of immune-related adverse events (irAEs) affecting various organs, including endocrine organs (thyroid, adrenal, pancreas, and pituitary glands), colon, lungs, skin, liver, kidneys, and heart. The severity of each irAE was graded according to the Common Terminology Criteria for Adverse Events (CTCAE) v5.0 classification [26]. Data were gathered through chart reviews, clinical notes, and radiological and laboratory reports.

### 2.4. Next-Generation Sequencing (NGS)

The assessment of tissue TMB was performed on archived or newly obtained formalin-fixed paraffin-embedded tumor samples using FDA-approved platforms, including the FoundationOne CDx assay (Foundation Medicine, Boston, MA, USA) and Tempus xT Gene Panel (Tempus, Chicago, IL, USA). The Guardant360 CDx assay (GuardantHealth, Palo Alto, CA, USA) also reported blood-based TMB. High TMB was defined as ≥10 mut/Mb, based on the cutoff established by the KEYNOTE-158 study and the FDA [16]. These platforms also provided tumor mutational profiling using NGS to detect genetic alterations present in tumor tissue. NGS platforms varied in sample type, gene coverage, and fusion detection. FoundationOne CDx utilized hybrid-capture targeted DNA sequencing on formalin-fixed paraffin-embedded (FFPE) tumor tissue. In addition to hybrid-capture targeted DNA sequencing, Tempus xT also included RNA sequencing for broader coverage. Guardant360 CDx utilized plasma samples to detect somatic mutations in circulating tumor DNA via hybrid-capture targeted DNA sequencing. Platform choice was left to clinician discretion. Mutations deemed pathogenic or likely pathogenic were used in the analysis.

### 2.5. Ethics Approval

The Northwestern University institutional review board approved the study protocol, which was conducted in compliance with Good Clinical Practice and the Declaration of Helsinki.

### 2.6. Statistical Analysis

The efficacy analysis population comprised all patients who received at least one dose of an ICI with evaluable TMB data and follow-up imaging for evaluation. The safety analysis population included all patients who received at least one dose of an ICI. A patient was considered not evaluable if a baseline assessment was obtained, but no post-baseline assessment had been performed. Descriptive statistics were used to summarize patient characteristics. Continuous variables were summarized using median and interquartile range (IQR) and compared between groups using the Wilcoxon rank sum test. Kaplan–Meier curves were used to estimate PFS and OS, and groups were compared using the log-rank test. Cox proportional hazard models were used to estimate hazard ratios (HR), and the proportional hazards assumption was checked [27]. Mutations identified by NGS were grouped into common signaling pathways responsible for carcinogenesis [28], and an oncoplot [29] was generated based on available co-mutation data. Unadjusted *p*-values are reported, with α < 0.05 considered statistically significant. All analyses were conducted in R Version 4.3.1.

## 3. Results

### 3.1. Patient Characteristics

Baseline characteristics, treatments administered, and immune biomarkers are summarized in Table 1.

In total, 117 patients were enrolled, including 105 (95%) patients evaluable by iRECIST criteria. The median age was 68, and 61% were male. The study population comprised 88% White, 6% African American, and 3.4% Asian participants. The cohort encompassed 14 different primary malignancies, including melanoma (33%; 39/117), NSCLC (28%; 33/117), SCLC (6%; 7/117), CRC (5.1%; 6/117), PBC (4.3%; 5/117), and others. Of the patients, 47% (55/117) were treatment-naïve, while the remaining 53% (62/117) had received prior systemic treatment. ICIs included pembrolizumab (37%), nivolumab (16%), ipilimumab plus nivolumab (16%), atezolizumab (11%), ipilimumab (1.7%), durvalumab (0.9%), and other (6%). A total of 60% of patients had TMB ≥ 15 mut/Mb. Median TMB level by tumor type is shown in Table 2.

A total of 88 patients (75%) were considered ICI-sensitive (Group 1), while 29 (25%) were considered non-ICI-sensitive (Group 2). Median TMB value for ICI-sensitive patients was 21.1 mut/Mb (IQR: 13.4–42.2 mut/Mb), compared to 15.0 mut/Mb (IQR: 11.4–22.8 mut/Mb) for non-ICI-sensitive patients. PD-L1 CPS expression level was positive in 46% and negative in 39% of patients.

At baseline, 31 patients (27%) had liver metastasis, while 83 patients (71%) did not. Data for three patients were missing or unknown. Of those in the ICI-sensitive group and the non-ICI-sensitive group, 67 (76%) and 16 (55%), respectively, did not present with liver metastasis.

irAEs were infrequent, with colitis (8%; 9/117), hepatitis (8%; 9/117), and skin reactions (6%; 7/117) being the most commonly reported (Table 3).

### 3.2. Gene Data

A total of 105 patients had NGS data, revealing 3899 different types of mutations. In descending order, frequency of mutations included missense mutations (80%, n = 2942), nonsense mutations (7.9%, n = 292), splice mutations (3.5%, n = 128), frameshift mutations (3.3%, n = 120), other mutations (2.5%, n = 931), substitutions (1.5, n = 54), and promoter/regulatory mutations (1.3%, n = 47).

A co-mutation plot, which summarizes all mutations according to response status, TMB, and type of mutation, is illustrated in Figure 1. The most common mutations included *TP53* (66.6%), *LRP1B* (40.9%), *TERT* (40%), *CDKN2A* (38%), *NF1* (36.1%), *MLL2* (35.2%), and *ROS1* (27.6%).

*TERT* and *CDKN2A* mutations were more frequently observed in ICI-sensitive tumors (*p* < 0.001 and *p* = 0.022, respectively), while *CDK12* mutations were more often found in non-ICI-sensitive tumors (*p* = 0.025). *TP53* mutations were significantly more prevalent in lung cancer compared to melanoma or other types of cancer (*p* < 0.001). *BRAF* mutations were more commonly identified in melanoma (*p* < 0.001).

### 3.3. Efficacy

Among all patients, 51% were responders, while 49% were non-responders. Of 105 evaluable patients, ORR was 34% across all malignancies (CR 14%). Within the ICI-sensitive subgroup, the ORR was 37%, compared to 27% in the non-ICI-sensitive group. Waterfall plots show the percentage change in tumor size from baseline for TMB and ICI-sensitive status (Figure 2).

The median PFS and OS were 8.05 months and 26.8 months, respectively (Figure 3). Median follow-up among patients who were censored for OS was 33 (IQR: 21–44) months.

Survival based on subgroup is shown in Figure 4. Those with ICI-sensitive tumors demonstrated improved PFS (*p* = 0.009) and OS (*p* = 0.014) compared to non-ICI-sensitive tumors (Figure 4A). Liver metastasis was associated with worse PFS (*p* = 0.015) and OS (*p* = 0.006) (Figure 4B). A cutoff of TMB ≥ 15 mut/Mb did not show significant difference in PFS (*p* = 0.95) or OS (*p* = 0.79) in the ICI-sensitive tumor subset; however, it was linked to improved PFS (*p* = 0.012) in patients with non-ICI-sensitive tumors (Figure 4C). Lack of previous systemic therapy also demonstrated improved PFS (*p* = 0.001) and OS (*p* < 0.001) (Figure 4D). Other factors such as age (*p* = 0.8), sex (*p* = 0.9), and PD-L1 status (*p* > 0.9) did not exhibit prognostic significance. Cox proportional hazards assumptions were satisfied for all predictors.

Mutations in the *MYC* pathway (*p* = 0.03) and *MLL2* (*p* = 0.014) were linked to a poorer response, whereas mutations in the *TERT* gene (*p* = 0.031) were associated with improved response. The following pathways did not show significant prognostic or predictive value for survival: *TP53* (*p* = 0.3), *RTK/RAS* (*p* > 0.9), *PIK3* (*p* = 0.4), *NOTCH* (*p* = 0.5), *WNT* (*p* = 0.9), *NRF2* (*p* = 0.2), *TGFβ* (*p* = 0.8), and *HIPPO* (*p* > 0.9) (Appendix A).

## 4. Discussion

In this work, the study presented in [30], is expended upon. We found that patients with high TMB had similar response rates to ICIs as reported in the KEYNOTE-158 trial. Type of tumor, absence of liver metastasis, lack of prior systemic therapy, and presence of *TERT* mutation were associated with improved responses to ICIs. Additionally, TMB ≥ 15 mut/Mb correlated with improved responses among patients categorized as non-ICI-sensitive. Conversely, liver metastasis and specific mutations in *MYC* and *MLL2* genes were associated with a poor response to ICIs across all patient groups. Factors such as PD-L1 status, age, and gender did not demonstrate significant prognostic value.

TMB has both limited sensitivity and specificity for the prediction of benefit from ICIs [31]. Previous studies examining the association of TMB with ICI responses in specific tumor types often failed to confirm the positive associations observed in more diverse cohorts [24,32]. Additionally, recent immunoscore trials suggest that a high TMB alone may not be a reliable predictor of responses to ICIs in solid tumors, suggesting potential limitations in the clinical utility of TMB [33]. Our study reinforces earlier studies highlighting the significance of tumor type when utilizing TMB-H as a predictive biomarker. We categorized patients with TMB-H into two groups based on their eligibility for standalone ICI treatment approved by the FDA: ICI-sensitive and non-ICI-sensitive [34]. The non-ICI-sensitive group primarily consisted of patients with CRC, PBC, and SCLC. In contrast, the ICI-sensitive group included patients with melanoma, NSCLC, urothelial cancer, and SCC. Notably, patients in the ICI-sensitive group experienced significantly better survival rates with ICI treatments.

The influence of tumor type on survival underscores the importance of various clinical and molecular factors. One significant clinical factor that may affect the efficacy of ICIs in solid tumors is the presence of liver metastases. Recent findings indicate that the immunosuppressive environment of the liver may contribute to resistance to ICIs among patients with liver metastases [35,36,37,38,39]. In our study, subgroup analysis revealed that 20% of patients in the ICI-sensitive group had liver metastases. In comparison, 45% of patients in the non-ICI-sensitive group had liver metastases. This higher incidence of liver metastases in the non-ICI-sensitive group may explain their reduced response to ICIs. Moreover, our study demonstrated a significant difference in median PFS and OS between patients with and without liver metastases, which aligns with recent data.

Numerous studies indicate that incorporating underlying mutational processes alongside traditional TMB may provide more accurate guidance for applying ICIs [40,41]. For instance, clinical studies involving the same tumor type have yielded differing results regarding patients classified as having high TMB. The phase I/II non-randomized trial, CheckMate 032, reported improvements in ORR and one-year OS rates in SCLC patients with high TMB, particularly among those receiving a combination of nivolumab and ipilimumab [42]. In contrast, the phase III trial, IMpower133, did not demonstrate an enhancement in OS for patients with high TMB who received atezolizumab combined with platinum-based chemotherapy compared to those with low TMB subjected to the same treatment regimen [43]. The high prevalence of the *MYC* oncogene in SCLC may account for this discrepancy [44], as elevated *MYC* expression has been associated with reduced survival following anti-PD-L1 treatment in solid tumors [45,46,47,48]. Our study has identified a correlation between *MYC* mutations and a low response rate to immunotherapy in solid tumors characterized by high TMB. *MYC* gene mutations have been associated with promoting tumor immune evasion through multiple mechanisms, including suppression of interferon signaling, downregulation of antigen presentation (such as MHC class I), upregulation of immune checkpoint molecules, and remodeling of the tumor microenvironment. These changes collectively result in a “cold” tumor immune phenotype with low T-cell infiltration, plausibly resulting in diminished response to immune checkpoint blockade [49].

MLL2 (also known as KMT2B) is a histone lysine methyltransferase that plays a crucial role as an epigenetic regulator of transcription [50]. *MLL2* alterations are present in some immunotherapy-responsive cancer types, such as up to 29% of melanoma [51]. In some cancers, such as early-stage lung SCC, cervical cancer, and gastrointestinal diffuse large B cell lymphoma, *MLL2* has been linked to poor prognosis [52,53,54,55]. The role of *MLL2* gene alterations as a biomarker for cancer immunotherapy is still being explored. A study involving various solid tumors found that in a small group of patients with *MLL2* deleterious alterations, there was an association with higher TMB and an improved survival after ICI therapy, though *MLL2* was not independently predictive of response to ICI [56]. A study assessing 11 long-term survivors with extensive-stage SCLC suggests that alterations in *MLL2* may predict better survival outcomes for patients undergoing first-line chemoimmunotherapy [57]. In this same study, in order to validate findings, a large pan-cancer immunotherapy cohort was assessed, and *KMT2B* mutations correlated with worse OS (*p* = 0.007). Moreover, *KMT2B* had divergent effects on immunotherapy response, with higher *KMT2B* expression associated with response in melanoma patients (*p* = 0.043) and lower *KMT2B* expression linked to response in stomach adenocarcinoma patients (*p* = 0.001). In our study, alterations in *MLL2* were overall associated with a poor response to ICIs in TMB-H solid tumors, suggesting that the response to ICI in the presence of *MLL2* alterations varies according to tumor type or *MLL2* alteration type. While the exact mechanism for this discrepancy is not known, the role of MLL2 in the tumor microenvironment continues to be elucidated. In lymphoma models, loss of function of *MLL2* is known to disrupt enhancer-mediated transcriptional regulation, leading to reduced expression of genes involved in antigen presentation and interferon signaling [58,59]. While this may not be applicable to carcinomas, it is possible that the resultant altered tumor immune microenvironment may promote immune evasion and explain suboptimal response to immune checkpoint inhibitor therapy in certain tumor types.

*TERT* mutations are significantly associated with increased TMB and a higher neoantigen load, which can lead to improved responses to ICIs [60]. Numerous studies have demonstrated that *TERT* mutations are linked to more favorable responses to ICIs in solid tumors [61]. However, a recent study found that *TERT* mutations are associated with poorer PFS in biliary tract carcinoma [62]. Additionally, co-mutations of *TP53* and *TERT* have been linked to worse survival outcomes in patients with hepatocellular carcinoma [63]. Unfortunately, we were unable to evaluate these co-mutation correlations in our cohort due to limited sample size. However, these findings emphasize that mutational profiles alone are insufficient for predicting ICI response, highlighting the importance of cancer type and co-occurring mutations.

In the Keynote-158 trial, response rates for tumors varied by TMB: 6.7% for TMBs < 10 mut/Mb, 12.5% for 10–13 mut/Mb, and 37% for >13 mut/Mb. This suggests the 10 mut/Mb cutoff is ineffective for distinguishing responders [16]. This threshold was established based on earlier retrospective studies involving patients with NSCLC [64]. However, numerous studies conducted since then suggest that a reevaluation of appropriate, potentially cancer-type-specific TMB cutoffs is necessary to improve patient selection for those most likely to benefit from ICIs [24,65]. In our study, 60% of patients demonstrated a TMB ≥ 15 mut/Mb. However, evaluating the entire patient population, this elevated TMB was not linked to enhanced survival outcomes. Conversely, within the subgroup of patients who were non-sensitive to ICIs, those with a TMB ≥ 15 mut/Mb exhibited significantly improved survival compared to individuals with TMB values ranging from 10 to 15 mut/Mb. The optimal TMB threshold remains unclear, though establishing one optimal cutoff in heterogenous tumor types with variable tumor microenvironments may not be possible. Further studies to identify optimal cutoffs, particularly focusing on specific tumor types, are warranted.

The limitations of this study include a single center, a small sample size, and its retrospective nature. As with other retrospective cohort studies, immortal time bias is a limitation, though statistical considerations were employed to attempt to mitigate this. Additionally, most patients had likely received ICIs for a different FDA-approved indication, which introduces the potential for confounding factors.

## 5. Conclusions

Our study highlights the predictive validity of TMB as a critical biomarker for assessing the effectiveness of ICIs across various cancer types. The benefits of TMB are especially significant in patients with *TERT* mutations and who do not have liver metastases. Additionally, TMB of ≥15 mut/Mb were associated with better outcomes for non-ICI-sensitive tumor types. On the other hand, we found that specific mutations, such as those in the *MYC*, and *MLL2* genes, can negatively impact the effectiveness of these treatments.

## Figures and Tables

**Figure 1 cancers-17-02673-f001:**
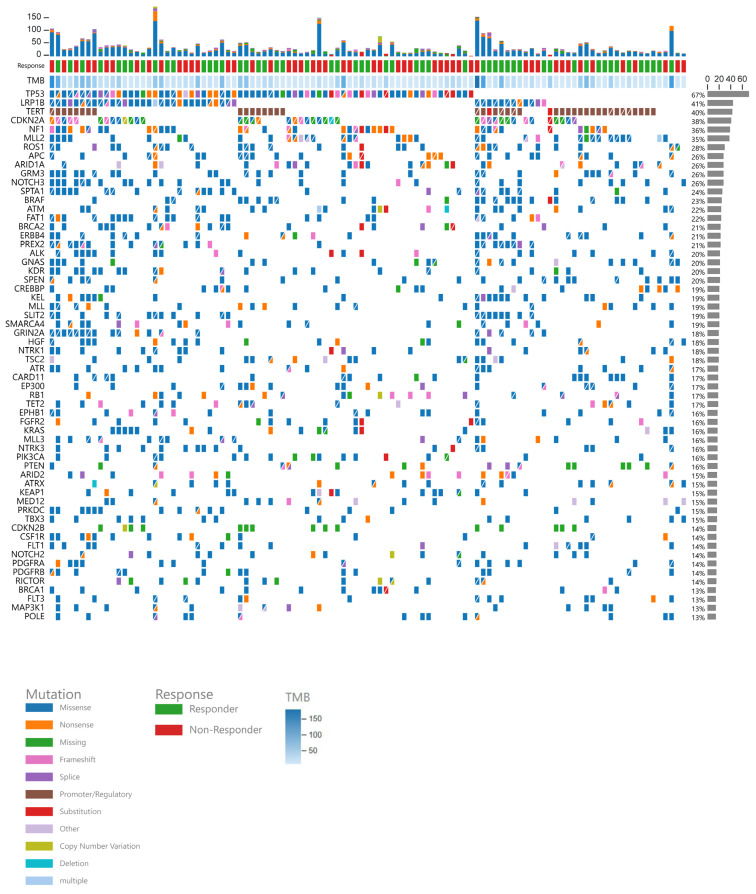
Co-mutation plot displaying mutations per patient (N = 105), with each patient’s mutations listed in individual columns. The stacked bar plot at the top shows the total number of mutations for each patient, followed by their response status, TMB level, and specific detected mutations. The bar graph on the right displays the percentage of patients with each mutation.

**Figure 2 cancers-17-02673-f002:**
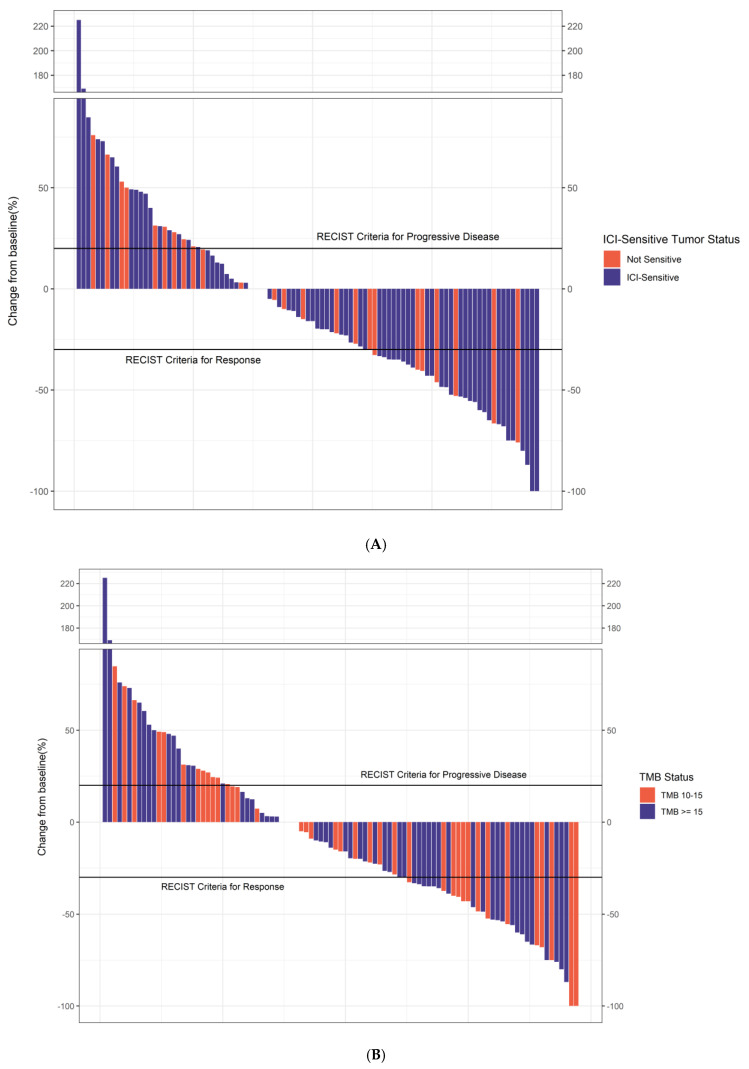
Waterfall plot showing the percentage change in tumor size from baseline based on ICI-sensitive status (**A**) and TMB status (**B**) (N = 117).

**Figure 3 cancers-17-02673-f003:**
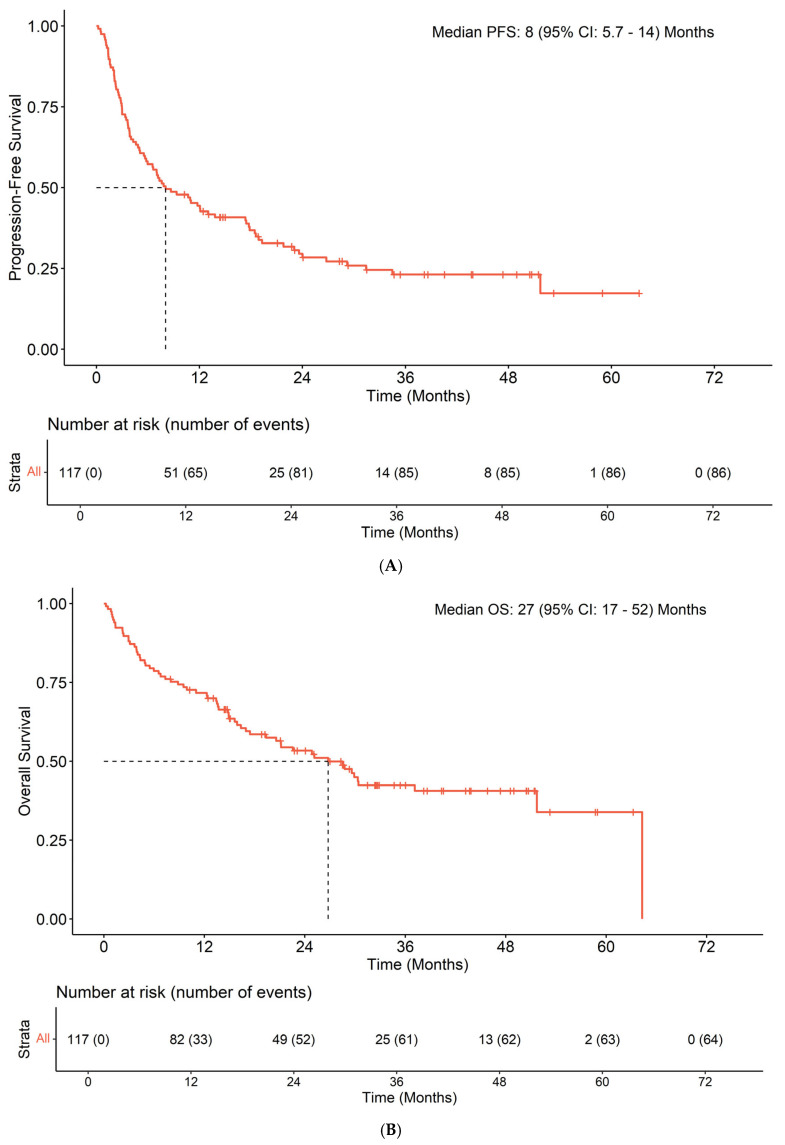
Kaplan–Meier estimates and risk tables for PFS (**A**) and OS (**B**) for the overall population (N = 117).

**Figure 4 cancers-17-02673-f004:**
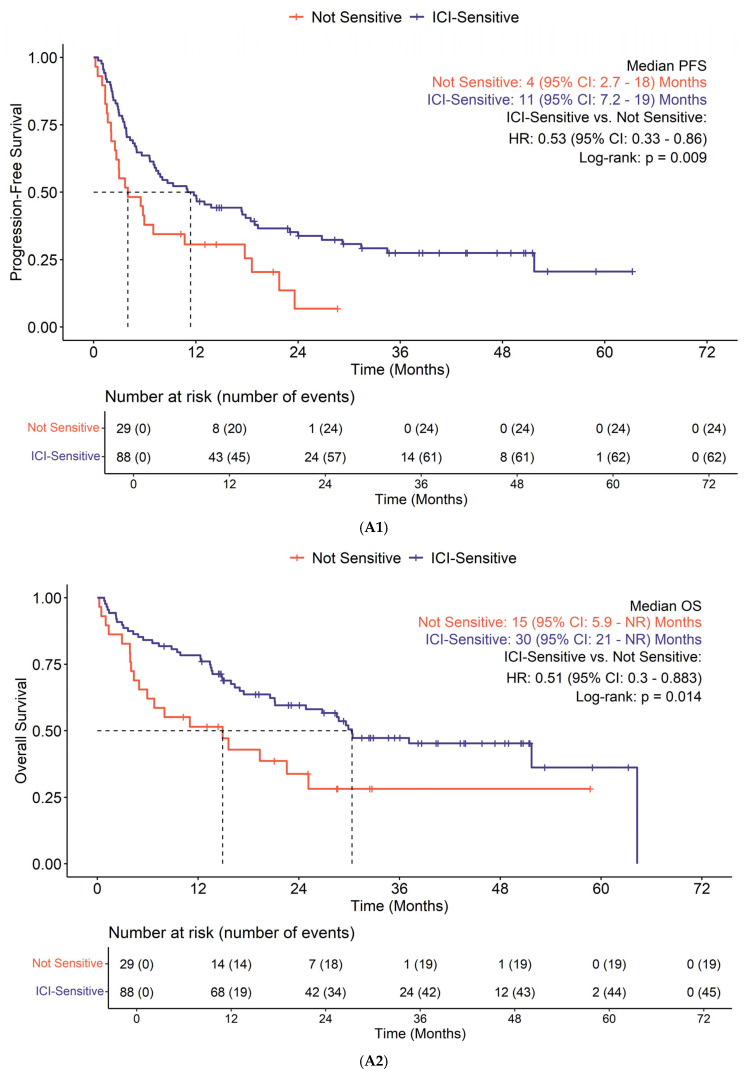
(**A1**): ICI-sensitive tumors demonstrated improved PFS compared to non-ICI-sensitive tumors. (**A2**): ICI-sensitive tumors demonstrated improved OS compared to non-ICI-sensitive tumors. (**B1**): Liver metastasis was associated with worse PFS. (**B2**): Liver metastasis was associated with worse OS. (**C1**): A cutoff of TMB ≥ 15 mut/Mb showed a significant difference in PFS in patients with non-ICI-sensitive tumors. (**C2**): A cutoff of TMB ≥ 15 mut/Mb did show significant difference in OS in patients with non-ICI-sensitive tumors. (**D1**): Lack of previous systemic therapy demonstrated improved PFS. (**D2**): Lack of previous systemic therapy demonstrated improved OS.

**Table 1 cancers-17-02673-t001:** Baseline characteristics in the overall group and based on treatment response.

Characteristic	Overall, n = 117 ^1^	[SD, <6 m]/PD, n = 51 ^1^	CR/PR/[SD, >6 m], n = 54 ^1^	*p*-Value ^2^
Age	68 (62, 76)	68 (62, 77)	69 (62, 76)	0.8
Sex				0.9
Female	46 (39%)	20 (39%)	22 (41%)	
Male	71 (61%)	31 (61%)	32 (59%)	
Race				0.2
Asian	4 (3.4%)	3 (5.9%)	1 (1.9%)	
Black/African American	7 (6.0%)	1 (2.0%)	4 (7.4%)	
Unknown	3 (2.6%)	2 (3.9%)	0 (0%)	
White	103 (88%)	45 (88%)	49 (91%)	
PD-L1 status				>0.9
Positive	36 (46%)	16 (46%)	18 (49%)	
Negative	31 (39%)	13 (37%)	13 (35%)	
NE	12 (15%)	6 (17%)	6 (16%)	
Unknown	38	16	17	
Previous therapies				0.039
No systemic therapy	55 (47%)	19 (37%)	31 (57%)	
Adjuvant/neoadjuvant/definitive therapy	62 (53%)	32 (63%)	23 (43%)	
Previous therapy lines				0.10
One line	29 (48%)	11 (35%)	14 (61%)	
Two lines	17 (28%)	10 (32%)	5 (22%)	
Three lines	5 (8.2%)	5 (16%)	0 (0%)	
Four or more lines	10 (16%)	5 (16%)	4 (17%)	
Unknown	56	20	31	
All responses				<0.001
Complete response	15 (13%)	0 (0%)	15 (28%)	
Partial response	21 (18%)	0 (0%)	21 (39%)	
Stable disease	22 (19%)	4 (7.8%)	18 (33%)	
Progressive disease	47 (40%)	47 (92%)	0 (0%)	
NE	8 (6.8%)	0 (0%)	0 (0%)	
N/A	4 (3.4%)	0 (0%)	0 (0%)	
First line immunotherapy				0.048
Ipilimumab	2 (1.7%)	2 (3.9%)	0 (0%)	
Nivolumab	32 (27%)	15 (29%)	14 (26%)	
Ipilimumab/Nivolumab	19 (16%)	5 (9.8%)	11 (20%)	
Pembrolizumab	43 (37%)	16 (31%)	23 (43%)	
Atezolizumab	13 (11%)	10 (20%)	2 (3.7%)	
Durvalumab	1 (0.9%)	0 (0%)	1 (1.9%)	
Other	7 (6.0%)	3 (5.9%)	3 (5.6%)	
Known ICI-sensitive tumor				0.13
Non-ICI-sensitive	29 (25%)	16 (31%)	10 (19%)	
ICI-sensitive	88 (75%)	35 (69%)	44 (81%)	
Tumor type				0.2
Adrenocortical carcinoma	2 (1.7%)	1 (2.0%)	1 (1.9%)	
Anal squamous cell carcinoma	2 (1.7%)	1 (2.0%)	0 (0%)	
Breast cancer	2 (1.7%)	0 (0%)	1 (1.9%)	
Colorectal carcinoma	6 (5.1%)	4 (7.8%)	2 (3.7%)	
Gynecologic carcinoma	4 (3.4%)	2 (3.9%)	2 (3.7%)	
Melanoma	39 (33%)	12 (24%)	24 (44%)	
Non-small cell lung carcinoma	33 (28%)	18 (35%)	12 (22%)	
Other skin	6 (5.1%)	1 (2.0%)	4 (7.4%)	
Pancreatobiliary	5 (4.3%)	2 (3.9%)	2 (3.7%)	
Small cell lung carcinoma	7 (6.0%)	3 (5.9%)	3 (5.6%)	
Unknown primary	4 (3.4%)	4 (7.8%)	0 (0%)	
Upper GI	2 (1.7%)	1 (2.0%)	1 (1.9%)	
Urothelial carcinoma	4 (3.4%)	2 (3.9%)	2 (3.7%)	
Head and neck SCC	1 (0.9%)	0(0%)	1(1.9%)	
MSI status				0.5
MSI stable	113 (96.6%)	49 (96%)	52 (96.3%)	
N/A	4 (3.4%)	2 (4%)	2 (3.7%)	
TMB status				0.5
TMB 10–15	47 (40%)	22 (43%)	20 (37%)	
TMB ≥ 15	70 (60%)	29 (57%)	34 (63%)	

^1^ Median (IQR); n (%). ^2^ Wilcoxon rank sum test; Pearson’s chi-squared test; Fisher’s exact test.

**Table 2 cancers-17-02673-t002:** Summary of TMB levels by tumor type.

Group	Tumor Type	Median TMB Level (mut/Mb)	IQR (mut/Mb)
ICI-sensitive	Unknown primary (SCC)	27.8	21.1, 49.5
	Melanoma	26.5	15.1, 56.7
	Non-small cell lung cancer	14.9	13.5, 26.5
	Adrenal corticoid carcinoma	35.8	29.8, 41.9
	Anal SCC	10.1	10.1, 10.1
	Head and neck SCC	16.7	16.7, 16.7
	Genitourinary carcinoma ^¥^	15.8	13.9, 19.5
	Other skin cancers *	27.7	13.9, 48.2
Non-ICI-sensitive	Breast cancer	11.3	10.9, 11.7
	Colorectal cancer	15.2	12.0, 27.8
	Gynecologic cancers (non-SCC)	11.3	10.3, 24.1
	Pancreaticobiliary cancers	15.1	15.0, 19.3
	Small cell lung cancer	14.9	11.0, 17.1
	Upper gastrointestinal cancers ^Ɨ^ (non-SCC)	15.3	13.3, 17.3

^¥^ Renal cell carcinomas and urothelial carcinomas; * basal cell carcinomas and squamous cell carcinomas; ^Ɨ^ esophageal, stomach, and duodenal adenocarcinomas.

**Table 3 cancers-17-02673-t003:** Immune-related adverse events classified by severity per CTCAE v5.0.

Adverse Event	Grade 1	Grade 2	Grade 3	Grade 4	Grade 5	Total (n = 48/117) (41%)
Thyroid disease	3 (2.5%)	2 (1.7%)	0	0	0	5 (4.2%)
Adrenal insufficiency	1 (0.8%)	1 (0.8%)	0	0	0	2 (1.7%)
Colitis	2 (1.7%)	1 (0.8%)	4 (3.4%)	2 (1.7%)	0	9 (7.6%)
Pneumonitis	0	1 (0.8%)	3 (2.5%)	1 (0.8%)	1 (0.8%)	6 (5%)
Infusion reactions	0	1 (0.8%)	0	0	0	1 (0.8%)
Severe skin reactions	5 (4.2%)	1 (0.8%)	1 (0.8%)	0	0	7 (5.9%)
Hepatitis	2 (1.7%)	1 (0.8%)	3 (2.5%)	3 (2.5%)	0	9 (7.6%)
Nephritis	0	1 (0.8%)	1 (0.8%)	0	0	2 (1.7%)
Hypophysitis	0	5 (4.2%)	0	0	0	5 (4.2%)
Myocarditis	0	0	2 (1.7%)	0	0	2 (1.7%)
Type 1 DM	0	0	0	0	0	0
Pancreatitis	0	0	0	0	0	0

## Data Availability

The data presented in this study are available on request from the corresponding author.

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
