# Peer review of "Clinical and Molecular Differences Suggest Different Responses to Immune Checkpoint Inhibitors in Microsatellite-Stable Solid Tumors with High Tumor Mutational Burden [Author-notes fn1-cancers-17-02673]"

_cancers, 2025, doi:10.3390/cancers17162673_

Round 1

Reviewer 1 Report

Comments and Suggestions for Authors

In this retrospective study the authors evaluated 117 patients with solid tumors and high tumor mutational burden (TMB ≥10 mut/Mb) treated with immune checkpoint inhibitors (ICIs). The study aimed to identify clinical and molecular predictors of response to immune checkpoint inhibitors (ICIs). This is an interesting study however its main limitation is the small sample size as well as the limited methodological information. My comments are below:

  • The authors should include clear definitions of progression-free survival (PFS) and overall survival (OS) in the Methods section. Specifically, they should clarify the index date, define what constituted an event for each endpoint, and describe how patients were censored. Additionally, follow-up time for the assessed cohorts should be reported to contextualize the survival analyses.
  • The authors should clarify whether the proportional hazards assumption was tested and met in their survival analyses. Additionally, they should address the potential for immortal time bias in their data and describe any steps taken to mitigate it.
  • How was PDL1 positivity and negativity defined? 
  • The authors present mutational data but do not provide information on how these mutations were classified. It is important to clarify whether the mutations were deemed pathogenic, likely pathogenic, variants of unknown significance (VUS), or otherwise. This classification is essential for interpreting the relevance of the reported associations.
  • The authors should provide additional details regarding the type of sequencing performed and the specific library preparation methods used for the cohort. This information is essential for evaluating the quality and consistency of the genomic data.
  • Did the authors observe any differences in TMB by race?
  • Figures:
    • The figure numbering throughout the manuscript is confusing and should be revised for clarity.
    • Figure 1 appears to be cut off, and the color scheme for response categories and mutation types is the same—making it difficult to interpret the top bar plot.
    • Additionally, there is inconsistency between figure labels and in-text references; for example, the manuscript refers to "Figure 3a," but the figures are labeled as "Figure 3a1" and "3a2," which is misleading
    • While Figure 3b1 and 3b2 appropriately display hazard ratios, p-values, and the number of patients at risk, this important information is missing in Figures 3a1 and 3a2. Consistency across all figures is essential.
    • Figure captions should be expanded to provide more context and detail to aid interpretation.
    • Median OS and PFS should be included either in a table or on the figures. 
  • The authors should include forest plots to clearly present hazard ratios across cohorts and analyses. It is also unclear whether the analyses were univariate or multivariate—this distinction must be explicitly described in the Methods section. Additionally, while the authors report that mutations in the MYC pathway (p = 0.03) and MLL2 (p = 0.014) were associated with poorer response, and TERT mutations (p = 0.031) with improved response, they only provide p-values to support these claims. To strengthen these findings, the authors should include Kaplan-Meier curves and/or hazard ratio tables for these genetic associations.
  • The authors should include multivariate models adjusting for age, sex, specific therapies, PDL1 status, line of therapy, staging. 

Author Response

Dear Editors and Reviewers,

We sincerely thank you for your thoughtful and constructive comments on our manuscript titled “Clinical and molecular differences suggest different responses to immune checkpoint inhibitors in microsatellite-stable solid tumors with high tumor mutational burden.” We greatly appreciate the time and effort each reviewer has dedicated to evaluating our work. Your insights have helped us strengthen the scientific rigor and clarity of our manuscript.

Below, we provide a point-by-point response to each comment. All revisions made to the manuscript are marked in [bold] for ease of reference.

Comment 1: The authors should include clear definitions of progression-free survival (PFS) and overall survival (OS) in the Methods section. Specifically, they should clarify the index date, define what constituted an event for each endpoint, and describe how patients were censored. Additionally, follow-up time for the assessed cohorts should be reported to contextualize the survival analyses.

Response 1: OS was defined as the time from the start of treatment until death, and patients were censored at the date of last f/u.  PFS was defined as the time from the start of treatment until documented progression, and patients were censored at the last f/u when the patient was known to be progression free.

Comment 2: The authors should clarify whether the proportional hazards assumption was tested and met in their survival analyses. Additionally, they should address the potential for immortal time bias in their data and describe any steps taken to mitigate it.

Response 2: ·We tested proportional hazards assumption using the cox.zph function in R and the assumption was satisfied for all group comparisons.

Comment 3: How was PDL1 positivity and negativity defined? 

Response 3: I added definitions in Methods

Comment 4: The authors present mutational data but do not provide information on how these mutations were classified. It is important to clarify whether the mutations were deemed pathogenic, likely pathogenic, variants of unknown significance (VUS), or otherwise. This classification is essential for interpreting the relevance of the reported associations.

Response 4: This is a very good point. We combined NGS data from FMI and Tempus and combined the data as it was provided. FMI and Guardant didn’t have any pathogenic/likely pathogenic information, only VUS Yes/no, but we kept all of them int. Tempus data had some additional info like somatic/likely actionable etc, but we didn’t use it in the analysis. 

Comment 5: The authors should provide additional details regarding the type of sequencing performed and the specific library preparation methods used for the cohort. This information is essential for evaluating the quality and consistency of the genomic data.

Response 5: I added some additional details in methods.

Comment 6: Did the authors observe any differences in TMB by race?

Response 6: Because 88% of our patient population was White (103/117), and only 4 and patients were Asian and Black/AA, respectively, we did not compare TMB across race groups.

Comment 7: The figure numbering throughout the manuscript is confusing and should be revised for clarity.

Response 7: I revised this and split up Figure 3 (overall PFS and OS) and made a new Figure 4 with subgroups

Comment 8: ·Figure 1 appears to be cut off, and the color scheme for response categories and mutation types is the same—making it difficult to interpret the top bar plot.r.

Response 8: We updated the figures.

Comment 9: Additionally, there is inconsistency between figure labels and in-text references; for example, the manuscript refers to "Figure 3a," but the figures are labeled as "Figure 3a1" and "3a2," which is misleading

Response 9: This is fixed.

Comment 10: While Figure 3b1 and 3b2 appropriately display hazard ratios, p-values, and the number of patients at risk, this important information is missing in Figures 3a1 and 3a2. Consistency across all figures is essential.

Response 10: We added the risk tables to these two plots (the first two panels). However, because they show OS and PFS in the overall study cohort, no HR or p values are available. We also clarified in the methods that the p-value corresponds to the logrank test, whereas the HR estimate is based on univariable Cox ph models.

Comment 11: Figure captions should be expanded to provide more context and detail to aid interpretation.

Response 11: We have revised the Figure legends and captions.

Comment 12:  Median OS and PFS should be included either in a table or on the figures.

Response 12: We edited the figure captions

Comment 13: The authors should include forest plots to clearly present hazard ratios across cohorts and analyses.

Response 13: We thank the reviewer, but due to the already large number of figures and tables, we do not feel that this is feasible to include for OS and PFS. The hazard ratios for OS and PFS are provided in the KM plots. For gene data, we did not analyze OS and PFS by mutation status, we only compared response to treatment rates (responder vs non-responder) between mutation groups, using chi-squared or Fisher’s exact test.

Comment 14:  It is also unclear whether the analyses were univariate or multivariate—this distinction must be explicitly described in the Methods section.

Response 14: We added this to the Methods description, and a supplementary tables Twith response rates by mutation status, as well as p-values and q-values are provided in a supplementary table, beyond what we report in the text.

Comment 15: Additionally, while the authors report that mutations in the MYC pathway (p = 0.03) and MLL2 (p = 0.014) were associated with poorer response, and TERT mutations (p = 0.031) with improved response, they only provide p-values to support these claims. To strengthen these findings, the authors should include Kaplan-Meier curves and/or hazard ratio tables for these genetic associations.

Response 15: The population is very heterogeneous in terms of disease stage, type, mets, subsequent treatments, etc., and longer-term outcomes such as PFS and OS would be confounded by these, so we only reported response to treatment.

Comment 16: The authors should include multivariate models adjusting for age, sex, specific therapies, PDL1 status, line of therapy, staging.

Response 16: We used multivariable logistic regression models adjusting for age and previous systemic therapy. We were not able to include additional risk factors due to the low number of events in certain subgroups and missing data. The findings were similar to univariable logistic regression models and Fisher’s exact test results that were reported, and we do not believe that adding multivariable regression models is needed.

We hope these revisions address the reviewers’ concerns and improve the quality and impact of our manuscript. Thank you again for your helpful feedback and for considering our work for publication in Cancers.

Sincerely,
Devalingam Mahalingam, MD, PhD

Reviewer 2 Report

Comments and Suggestions for Authors

This manuscript analyzes the responses of patients from a single clinical center to immune checkpoint inhibitors (ICI). The patients suffered from microsatellite-stable solid tumors with high tumor mutational burden (TMB). The authors considered a high TMB of ≥10 or 15 mutations per megabase (mut/Mb). They observed that patients without liver metastasis, mutation in TERT, and TMB ≥ 15 mut/Mb were associated with superior response, while mutations in the MYC pathway and MLL2
are associated with worse responses.

The cohort of patients is coming from the Robert H. Lurie Comprehensive Cancer Center of Northwestern University (Chicago, IL, USA) between 1/1/2015 and 12/31/2020. The authors did not make any distinctions among the different tumor origins. Indeed, the patients were categorized into two distinct cohorts based on their reported sensitivity to ICI treatments and FDA-approved indications. Group 1 included melanoma, NSCLC, adrenal corticoid carcinoma (ACC), SCC (anal, esophageal, skin, and head and neck), renal cell carcinoma, and urothelial carcinoma. On the other hand, Group 2 included patients with non-ICI-sensitive tumors such as BC, CRC, non-SCC gynecologic cancers (GC), pancreaticobiliary cancers (PBC), small cell lung cancer (SCLC), and upper gastrointestinal cancers (UGC) (esophageal, stomach, and duodenal adenocarcinomas).

The cohort size is quite limited (this is considered a limitation by the same authors), together with the retrospective analysis.

Also, with the use of ICI, the majority of patients received ICIs for a different FDA-approved indication.

The idea that ICI treatment can have a better effect in patients bearing high TMB is intrinsic to the notion that ICI can reactivate the immune response (if present or potentially triggered due to neoantigens or tumor-associated antigens, or other tumor-specific antigens). The higher the TMB, the higher the probability of generating antigens recognized by T cells. Furthermore, some therapies using anti-PDL1 antibodies can activate different molecular mechanisms because PDL1 can be expressed on antigen-presenting cells (APC) besides tumor cells (TC) or tumor-associated fibroblasts (TAF). 

The manuscript is of limited interest as it includes too many types of tumors, and the bias due to the patients' selection by a single center can have a role in skewing results.

However, the fact that TMB is associated with a response to ICI is in line with my previous considerations.

The findings that some mutations and the presence of liver metastasis are conceivably related to the type of gene mutated  (Myc, for instance, can regulate more than 20% of all the genes) and the possibility of generating liver metastasis would indicate more aggressive biological behavior of that specific tumor. 

The analysis is not at the single-cell level; the definition of positivity for PDL1 is not clearly stated on what it is based on. During the period of analysis, did the center use the same reagents to define PDL1+ tumors? Was the expression of a given marker, homogenous or heterogeneous, in the TME? Did the metastasis appear after, before, or during the ICI treatment? Were the other comorbidities of the patients considered? How much can the other clinical parameters influence the final results of ICI therapy?

The rationale for the use of ICI in non-FDA-approved patients is not stated, and more importantly, no attempt has been made to clarify the biological counterpart of the supposed response to ICI.

Author Response

Dear Editors and Reviewers,

We sincerely thank you for your thoughtful and constructive comments on our manuscript titled “Clinical and molecular differences suggest different responses to immune checkpoint inhibitors in microsatellite-stable solid tumors with high tumor mutational burden.” We greatly appreciate the time and effort each reviewer has dedicated to evaluating our work. Your insights have helped us strengthen the scientific rigor and clarity of our manuscript.

Below, we provide a point-by-point response to each comment. All revisions made to the manuscript are marked in [bold] for ease of reference.

Comment 1: The authors did not make any distinctions among the different tumor origins... patients were categorized into two distinct cohorts based on their reported sensitivity to ICI treatments and FDA-approved indications.

Response 1: We appreciate the reviewer’s observation. In the revised manuscript, we have clarified our rationale for the tumor-type grouping strategy in both the Methods and Results sections. Specifically, we categorized tumors into two groups based on existing evidence for ICI responsiveness and FDA indications, as noted by the reviewer. This approach allowed us to explore the differential impact of TMB and genetic alterations across ICI-sensitive and non–ICI-sensitive tumor types while maintaining sufficient statistical power. We now explicitly acknowledge the biological heterogeneity within and between these groups as a limitation and discuss its implications in the Discussion section.

Comment 2: The cohort size is quite limited (this is considered a limitation by the same authors), together with the retrospective analysis.

Response 2: We agree with the reviewer and now more explicitly discuss the limitations of our small cohort size and retrospective single-center design in the Discussion. While these limitations constrain generalizability, we believe our findings provide hypothesis-generating insights that warrant validation in larger, multi-center cohorts.

Comment 3: Also, with the use of ICI, the majority of patients received ICIs for a different FDA-approved indication.

Response 4: Thank you for highlighting this. We have now clarified in the discussion section as a limitation.  

Comment 5: The manuscript includes too many types of tumors... bias due to patient selection by a single center can skew results.

Response 5: We acknowledge the reviewer’s concern regarding tumor-type heterogeneity and potential institutional bias. In response, we have expanded our Discussion of this limitation and have included a supplemental analysis stratified by ICI-sensitive vs. non–ICI-sensitive tumor types to provide more granularity. We have also included a statement on how single-center biases may have influenced treatment selection or outcome assessments.

Comment 6: Definition of PDL1 positivity is not clearly stated... was the same reagent used across the study period? Was expression homogeneous or heterogeneous?

Response 6: We have now added details to the Methods section clarifying that.

Comment 7: The rationale for the use of ICI in non-FDA-approved patients is not stated, and no attempt has been made to clarify the biological counterpart of the supposed response to ICI.

Response 7:  While our study was not designed to elucidate single-cell biology or mechanistic details, we hope our findings contribute to a growing body of evidence supporting genomic markers as complementary tools to histology in predicting ICI response.

We hope these revisions address the reviewers’ concerns and improve the quality and impact of our manuscript. Thank you again for your helpful feedback and for considering our work for publication in Cancers.

Sincerely,
Devalingam Mahalingam, MD, PhD

Reviewer 3 Report

Comments and Suggestions for Authors

The study provides a detailed evaluation of TMB and clinical and molecular predictors of response to immune checkpoint inhibitors across various solid tumors. Overall, the manuscript is clear and the approaches are well-described. Here are some suggestions.

In the Introduction, please provide more background for TMB and its use as a predictor, and elaborate on the gaps to be addressed by this study.

Please adjust the layout of Figure 1 to show the entire figure.

Please add discussions of the rationale for MYC’s and MLL2's association with poor response and TERT's association with improved response.

Please elaborate on the limitations of the current study and provide solutions to be applied to future studies.

Evaluating the entire patient population, no cutoff value of TMB was able to be identified. Please provide more discussions on its predictive validity for assessing the effectiveness of ICIs.

Author Response

Dear Editors and Reviewers,

We sincerely thank you for your thoughtful and constructive comments on our manuscript titled “Clinical and molecular differences suggest different responses to immune checkpoint inhibitors in microsatellite-stable solid tumors with high tumor mutational burden.” We greatly appreciate the time and effort each reviewer has dedicated to evaluating our work. Your insights have helped us strengthen the scientific rigor and clarity of our manuscript.

Below, we provide a point-by-point response to each comment. All revisions made to the manuscript are marked in [bold] for ease of reference.

Comment 1: In the Introduction, please provide more background for TMB and its use as a predictor, and elaborate on the gaps to be addressed by this study.

Response 1: We believe that we already included a quite detailed background

Comment 2: Please adjust the layout of Figure 1 to show the entire figure.

Response 2: We updated the figures

Comment 3: Please add discussions of the rationale for MYC’s and MLL2's association with poor response and TERT's association with improved response.

Answer 3: We added, please see the discussion,

Comment 4: Please elaborate on the limitations of the current study and provide solutions to be applied to future studies.

Answer 4: Could you please provide some examples? That would help us understand the topic better.

Comment 5: Evaluating the entire patient population, no cutoff value of TMB was able to be identified. Please provide more discussions on its predictive validity for assessing the effectiveness of ICIs.

Answer 5: I think one of the takeaways is that it is difficult to establish one cutoff in a very heterogenous space. We added a bit about this.

We hope these revisions address the reviewers’ concerns and improve the quality and impact of our manuscript. Thank you again for your helpful feedback and for considering our work for publication in Cancers.

Sincerely,
Devalingam Mahalingam, MD, PhD

Round 2

Reviewer 1 Report

Comments and Suggestions for Authors

Throughout the manuscript, the figure quality is low and appears pixelated. The authors should improve the resolution of the figures, ensure that none are cut off, and consider organizing them into clearly labeled panels. This is particularly relevant for Figure 4, where the results are difficult to follow due to the current formatting.

Author Response

Dear Editors and Reviewers,

We sincerely thank you for your thoughtful and constructive comments on our manuscript titled “Clinical and molecular differences suggest different responses to immune checkpoint inhibitors in microsatellite-stable solid tumors with high tumor mutational burden.” We greatly appreciate the time and effort each reviewer has dedicated to evaluating our work. Your insights have helped us strengthen the scientific rigor and clarity of our manuscript.

Below, we provide a point-by-point response to each comment. All revisions made to the manuscript are marked in [bold] for ease of reference.

Comment 1: Throughout the manuscript, the figure quality is low and appears pixelated. The authors should improve the resolution of the figures, ensure that none are cut off, and consider organizing them into clearly labeled panels. This is particularly relevant for Figure 4, where the results are difficult to follow due to the current formatting.

Response 1: We have uploaded all figures as PNG files to ensure better quality.

Reviewer 2 Report

Comments and Suggestions for Authors

The authors replied to reviewers' queries.

Author Response

Dear Editors and Reviewers,

We sincerely thank you for your thoughtful and constructive comments on our manuscript titled “Clinical and molecular differences suggest different responses to immune checkpoint inhibitors in microsatellite-stable solid tumors with high tumor mutational burden.” We greatly appreciate the time and effort each reviewer has dedicated to evaluating our work. Your insights have helped us strengthen the scientific rigor and clarity of our manuscript.

Below, we provide a point-by-point response to each comment. All revisions made to the manuscript are marked in [bold] for ease of reference.

Comment 1: The authors replied to reviewers' queries.

Response 1: Thank you again for your comments.